# What can drawings tell us about children's perceptions of nature?

Kate Howlett●*◉, Edgar C. Turner●◉

Department of Zoology, University of Cambridge, Cambridge, Cambridgeshire, United Kingdom

◉ These authors contributed equally to this work.

* kh557@cam.ac.uk

**Data Availability Statement:** Data are archived on the Dryad Digital Repository: https://doi.org/10.5061/dryad.x3ffbg7pg.

**Funding:** K.H. is funded by the Natural Environment Research Council (grant number NE/

## Abstract

The growing disconnect between children and nature has led to concerns around the loss of ecological knowledge and reduced nature connection. Understanding children's perceptions of nature is vital for engaging them with local wildlife and mitigating this growing disconnect. This study investigated children's perceptions of nature by analysing 401 drawings made by children (aged 7–11) of their local green spaces, collected from 12 different English schools, including state-funded and privately funded. We assessed which animal and plant groups were drawn the most and least often, quantified each drawing's species richness and community composition, and identified all terms used in the drawings to the highest taxonomic resolution possible. The most commonly drawn groups were mammals (80.5% of drawings) and birds (68.6% of drawings), while herpetofauna were the least commonly drawn (15.7% of drawings). Despite not explicitly being asked about plants, 91.3% of drawings contained a plant. Taxonomic resolution was highest for mammals and birds, with 90% of domestic mammals and 69.6% of garden birds identifiable to species, compared to 18.5% of insects and 14.3% of herpetofauna. No invertebrates other than insects were identifiable to species. Within plants, trees and crops were the most identifiable to species, at 52.6% and 25% of terms respectively. Drawings from state-school children had higher plant richness than those from private-school children. Animal community composition differed between school funding types, with more types of garden birds drawn by private-school than state-school children, and more types of invertebrates drawn by state-school than private-school children. Our findings indicate that children's perceptions of local wildlife are focused on mammals and birds. While plants feature prominently, plant knowledge is less specific than animal knowledge. We suggest that this skew in children's ecological awareness be addressed through better integration of ecology within national curricula and more funding for green space within schools.

## 1. Introduction

The growing disconnect between people and nature has often been linked to rapid urbanisation and reduced daily contact with nature, resulting in a reduction of interactions and experiences both in childhood and across lifetimes [1–3]. In the UK, accessibility of green space is

L002507/1): https://www.ukri.org/councils/nerc/. The funder had no role in study design, data collection and analysis, decision to publish, or preparation of the manuscript.

**Competing interests:** The authors have declared that no competing interests exist.

highly heterogeneous [4–6], varying by socioeconomic background and distance from the home [7–9]. In addition, green space accessibility is lower for children than adults because of extra barriers, such as parental restrictions on independent movement, in part driven by parental concerns around safety [10], and urban barriers, such as roads [11–14]. Across Europe and North America, the area across which children regularly and independently move around the home has declined in recent decades [15–19]. As a result, a large proportion of green space is now inaccessible to children, even if physically present near the home [20].

Linked with this, frequent concerns have been raised around the 'extinction of experience', where children's opportunities to experience and develop a connection with the natural world are increasingly limited [21], with impacts on awareness. For example, children in the UK show greater familiarity with science fiction characters than with the commonest native species [22, 23], demonstrating that, while they have a large capacity for identification, children can be swayed more by images in the media than by those in the natural world [24]. A related concept is the 'transformation of experience', where direct experiences of nature are not just removed, but replaced by indirect experiences [25, 26]. These indirect experiences can be incidental or vicarious [27], and include technology-mediated experiences, such as through video games, social media, nature documentaries or other forms of entertainment. Although there is evidence that environmental knowledge, such as species identification and knowledge of species interactions, can be gained from these indirect experiences, even when the technology is not primarily designed for this purpose [28], there is not yet a good understanding of how indirect experiences facilitate a connection with nature [25, 26], by which we mean an individual's relationship with nature on a physical, emotional, spiritual or intellectual level [3, 29]. However, there is evidence that the frequency of indirect experiences, as well as of direct ones, influences children's attitudes towards biodiversity and their willingness to conserve it [30].

So far, there has been a degree of inconsistency over what constitutes 'indirect' nature experiences [3]; some argue that nature experiences cover a spectrum, from direct to vicarious interactions with nature, which occur when there is a lack of sensory contact and the interaction is instead technology-mediated. In this case, nature experiences are not necessarily reduced or 'extinct', but transformed. However, others argue that the confounding of direct and indirect nature experiences is not what was originally intended by the concept of 'extinction of experience', and that this term is better reserved for the loss of direct experiences with nature [3]. These two viewpoints are not incompatible: indirect experiences, such as nature documentaries, can be viewed both as part of the trend of the transformation of experience and as concurrent with the extinction of experience. Regardless, indirect experiences are today forming a greater part of children's interactions with nature, and these can be both positive or negative, incidental or intended, just as direct experiences can be [31].

More residential green space around a child's home has been linked with improved cognitive functioning, including across measures of intellectual, emotional and behavioural development [32, 33], and reduced psychological impacts of stressful life events [34], both when based on remote imaging of land-cover and on the 'naturalness' of the home environment (e.g., views of trees through a window). Local green spaces, such as gardens, local parks and school grounds are therefore likely to be important for delivering the wellbeing-related benefits of biodiversity. Indeed, the volume of vegetation in children's school grounds has been found to be associated with the perceived restorativeness of these spaces [35], and newly renovated green schoolyards in urban areas have been shown to increase children's positive social interactions [36], attentional development [37] and levels of physical activity, particularly for girls, as well as their appreciation of these spaces [38] and the development of resilience [39].

A reduced connection with nature is also likely to have detrimental effects on future support for conservation. Although vicarious nature experiences have been shown to influence

attitudes to biodiversity conservation [30], development of nature connection is highly dependent on direct experiences in the natural world [40–42]. While artificial exposure has been shown to increase pro-nature conservation behaviours in adults, such as monetary donations to animal protection organisations, it has not been linked to an increase in nature connection [43]. Direct experiences during childhood have a particularly significant influence on environmental attitudes later in life, such as recycling, beach clean-ups and political activity [44], as well as affecting long-term cognitive development [45, 46]. Raising support for conservation initiatives and recruitment of the next generation of naturalists and conservationists is therefore likely to become more of a challenge as nature disconnect grows and direct experiences in nature are replaced by indirect experiences. The activities encouraged and facilitated in local green spaces are therefore likely to be particularly important for building children's connection with nature, with both immediate effects and ramifications in later life.

Quantitative social research with child respondents is less common than with adults [47]. This is partly due to concerns over obtaining reliable information from children, linked to issues with children's understanding of what is being asked of them and retaining attention throughout the process [47, 48]. However, research shows that respondents as young as five years old can provide reliable information when they are asked in ways they can understand and respond to appropriately [48–51]. In particular, approaches using pictures and drawings have yielded detailed information on children's knowledge, feelings and attitudes [48–50, 52], including on ecological questions. For example, one UK study that asked children to draw a rainforest [53] found that children have a detailed view of rainforest environments that encompasses a diverse range of habitats and animal species, but that vertebrate taxa, especially mammals, birds and reptiles, were overrepresented, while invertebrate taxa, especially insects and worms, were underrepresented in comparison to their actual contributions to biomass and species richness [53]. A similar result was found by another UK study [52], in which children were asked to draw what they thought was in their school grounds before and after taking part in a nature engagement programme across the school year. While at first children drew more vertebrate than invertebrate species, their perception was more reflective of reality at the end of the engagement programme, when they drew more invertebrates [52].

More generally, drawings have been used to assess children's perceptions and meanings of 'nature' in an abstract sense [54], to assess the development of children's positive attitudes to small animals and invertebrates [55], to investigate how children perceive the functioning of the natural, geological and anthropic aspects of the environment [56], to explore environmental indictors of children's wellbeing, such as safety [57], and to evaluate children's biological understanding of animals as pets [58]. In our study, we used drawings to gauge children's awareness of local wildlife. Despite there being questions around the importance of ecological knowledge, such as species identification, for the development of nature connection [59], familiarity for particular taxa can engender positive attitudes [60] and a greater likelihood to think of a species as requiring conservation [61], which has implications for both policy and scientific research [62]. We therefore argue that gauging children's ecological knowledge in this way also has implications for nature connection and conservation.

In this study, we asked children (all aged between 7 and 11 years) from 12 different primary schools across England to draw the local wildlife living in their garden or local green space. We quantified animal and plant species richness and community composition of the drawings, as well as the taxonomic level to which drawings and associated labels could be identified. We also estimated the amount of green space to which children are regularly exposed on a daily basis by calculating the total area of green space in a 3km buffer around their school. We then assessed whether level of representation and identification differed between taxa, and whether there were differences in representations between state-funded and non-state-funded school

pupils, and across schools with differing levels of greenness of their surroundings. Specifically, we asked the following questions:

1. What are the most common types of animals and plants drawn, and does the level of taxonomic resolution change between taxa?

2. Does species richness, community composition and level of taxonomic resolution of animals and plants change between drawings from different school types or with greenness of school surroundings?

We expected that vertebrates, specifically mammals and birds, would be the most commonly drawn and identified to the highest taxonomic resolution. Secondly, we expected that both species richness and resolution of taxonomic identification would be higher in drawings from schools with greener surroundings. We interpret our findings in the context of their potential implications for future connection with nature.

## 2. Methods

### 2.1. Data collection

Drawings were collected from 401 primary-school children, aged between 7 and 11, at 12 different primary schools, distributed across England but with the highest concentration around the southeast (S1 Fig). Using the University Museum of Zoology, Cambridge (UMZC)'s mailing lists and social media accounts, we sent out information about the study, asking whether any primary school teachers would be happy to be contacted with further information. The mailing lists are specifically for primary-school teachers and are free for anyone to sign up to via the Museum's website. In total, 79 primary-school teachers at 57 different English primary schools said they were happy to be contacted with further information about the study. We arranged for children at as many of these schools as possible to provide drawings for us, limited by the teachers' and schools' capacity to support us. This totalled 401 children across 12 schools, including three different 'school types', that varied by fee-paying status and governance: three private, four academy and five state schools (S1 Table).

We split state-funded schools into two categories, state and academy, since they reflect different management practices. Although they are both free for children to attend, academies are administratively free from local-authority control, whilst state schools are administered by their local authority with regards to admissions and day-to-day running. Private schools are paid for by parents and are not subject to local-authority control. This means that state schools receive their funding from local authorities and are constrained by them in how budgets are spent, while academies receive funding directly from central government and have more freedom over spending. Private schools have complete freedom over how they spend their budgets. Given that outside space in schools, including green space, is not held to any particular standards by the Department for Education in the UK and budgets vary between school types, it is possible that the amount of money allocated to and availability of outside space varies between school types. In addition, given the fee-paying status of private schools and the subsequently greater proportion of children in higher income brackets attending private than state-funded schools, it is possible that privately educated children have greater access to private, domestic gardens. Collectively, these differences might influence daily exposure to nature for children attending different school types and consequently on awareness of local wildlife.

Children were provided with a worksheet (S1 Appendix) and the following instruction: 'Please draw and label a picture of your garden or local park showing the animals you think live there'. The worksheet also provided space for children to write a caption beneath their

drawing. We asked children about the green space they were exposed to outside school rather than about school grounds specifically to allow them to express their wider ecological knowledge, without being constrained by specific characteristics of their school. We specified local spaces to avoid children viewing the activity as a species-listing exercise and including all species they were aware of, regardless of whether or not they thought of them as local wildlife (e.g., elephants). The worksheet was emailed to teachers, who were asked to print them out, distribute them to their class during a spare ten minutes in the school day and to read the instruction at the top of the worksheet out loud. Teachers were asked not to give any extra information or guidance to children other than the written instruction.

## 2.2. Research ethics

Written consent was obtained from parents or guardians of all children who provided a drawing through a letter sent home via schools. Parents or guardians were provided with full Participant Information before being asked to provide written consent (S2 Appendix). Participation was voluntary, and it was made clear to parents or guardians that they were under no obligation for their child to take part and that they could withdraw their consent at any point (up to three months after the end of the study) with no penalties. Our protocol was reviewed and approved by the Cambridge Psychology Research Ethics Committee (PRE.2019.009).

## 2.3. Data processing

At the first stage of processing, we made a comprehensive list of all terms used in the labels and captions to refer to animals ($n$ = 123) and plants ($n$ = 56) (S2 Table). We decided to include plants in our analysis at this stage, despite not specifically asking children about plants, because they appeared in the majority of drawings, indicating that children connected plants with their local green spaces and therefore that they merited analysis. At the second stage of processing, we went back through each individual drawing to record whether each term was present or absent. Where a term was misspelled or hard to read, we marked it as present only if it was independently interpreted in the same way by a second member of the research team.

Where a drawing was not labelled, but the broad identity of the animal or plant was clear after independent agreement by a second researcher, we marked the most specific term possible as present (e.g., where a tree or flower was drawn but not labelled, 'tree' or 'flower' was marked as present). Where an unlabelled drawing had an unambiguous, diagnostic detail (e.g., a bird was drawn with a brown body and red breast, or an invertebrate was drawn with eight legs or dangling from a web), we afforded it a greater level of taxonomic identification (e.g., 'robin' rather than 'bird', or 'spider' rather than 'minibeast'), again following independent agreement by a second researcher, and marked this term as present. We then identified every term to the greatest taxonomic resolution possible (Species, Genus, Family, Order, Class, Phylum or Kingdom). Where a term had no taxonomic meaning (e.g., 'tree', 'blossom'), we recorded 'None' (S2 Table). We recorded the total number of different terms per drawing as the 'species richness', which we kept separate for animals and plants. We decided on the term 'species richness' since this is understandable by a broad audience, rather than because we wish it to be interpreted in a strictly ecological sense.

For the purposes of analysing community composition, we grouped all animal terms into one of the following categories: General (broad terms for groups of animals, e.g., 'Mammals', 'Birds', 'Creepy crawlies'), Domestic Mammals, Wild Mammals, Garden Birds (species regularly attracted to feed or nest in domestic gardens, e.g., 'Robin', 'Blue tit', 'Magpie'), Other Birds (species just as likely or more likely to be seen in parks or nature reserves than in a domestic garden, e.g., 'Duck, 'Swift', 'Red kite'), Herpetofauna, Insects, Other Invertebrates.

We decided to separate mammals into 'Domestic Mammals' and 'Wild Mammals' because domestic species featured in such a high proportion of drawings and in such high diversity (i.e., a high number of different domestic animals per drawing). As such, this distinction allows us to explore whether any trends are true across all mammals, or due more or less to either domestic or wild mammals. Similarly, we grouped all plant terms into one of the following categories: General (broad terms for groups of plants without taxonomic meaning, e.g., 'Trees', 'Flowers', 'Plants'), Trees, Flowers, Crops (plants grown for produce, e.g., Strawberries, Potatoes, Carrots), Other Plants ('Mushroom' is included in 'Other Plants' since it is the only fungal representative, and it appeared in just three drawings) (see S2 Table for a full list of terms in each category). Terms classified in the 'General' categories were not counted towards species richness, nor were the terms 'Herbs' or 'Vegetables' (in the 'Crops' category) since these do not have taxonomic meaning (S2 Table). The purpose of creating 'General' categories for both animals and plants was to remove terms with too low a level of taxonomic meaning to justify inclusion in the species richness total (e.g., 'Creepy crawlies' or 'Plant'), or were too imprecise to justify inclusion in one of the other categories for community composition analyses (e.g., 'Bugs' or 'Blossom').

To test for possible effects of the greenness of children's surroundings on perceptions of local wildlife, we calculated the area of green space in a 3km buffer around each school. We chose 3km since this is a good approximation of the average distance travelled to school by primary school children in the UK [63, 64], making this a reasonable representation of the greenness to which children are exposed on a regular basis. We first calculated the area of this space in hectares using QGIS [65] and imagery from Google Earth. We then used R version 4.1.3 [66] and R Studio Build 461 [67] to calculate the total number of pixels and number of green pixels in this area, which allowed us to calculate the area of green space (ha) in the buffer (hereafter referred to as 'buffer greenness').

## 2.4. Statistical analyses

All statistical analyses were performed in R version 4.1.3 [66] within R Studio version Build 461 [67]. We used *tidyr* [68], *RColorBrewer* [69], *ggsignif* [70], *ggplot2* [71] and *cowplot* [72] for data wrangling, exploration and visualisation. Exploration followed Zuur et al., 2010 [73]. We fitted generalized linear mixed models (GLMMs) using *glmmTMB* [74], multivariate generalized linear models (mGLMs) using *mvabund* [75], and chi-square tests of independence using *stats* [66]. Unless otherwise stated, we fitted models to negative binomial distributions, including school type (levels: state, academy, private) and buffer greenness as fixed effects and, for mixed models, school identity as a random intercept effect.

**2.4.1. Species richness.**   We analysed animal and plant species richness using GLMMs to account for non-independence of drawings from the same school. We fitted the models for plant species richness to a Poisson distribution, since those fitted to a negative binomial distribution displayed model convergence issues.

We validated GLMMs by plotting quantile residuals against predicted values and covariate school type to verify that no patterns were present. To ensure our GLMMs fitted the observed data, we ran simulation-based dispersion tests using *DHARMa* [76] to compare the variance of the observed residuals against the variance of the simulated residuals, with variances scaled to the mean simulated variance, and checked that our model was within the range of our simulations [77]. Our simulations indicated that there were no issues in model fit. We determined the significance of school type to each model by comparing fitted models with null models using likelihood ratio tests (LRTs). If mixed models suggested a moderately significant effect of school type ($0.03 < p < 0.07$), we re-calculated $p$ values based on parametric bootstrapping

using *DHARMa* [76, 78]. If school type was significant, we used *multcomp* [79] to conduct post-hoc analyses (Tukey all-pair comparisons, adjusting *p* values using the Bonferroni correction) to identify school types between which significant differences occurred.

**2.4.2. Community composition.** We analysed community composition at the category level, taking the number of terms used per category per drawing as 'abundance'. We used mGLMs [80, 81] to analyse animal and plant community composition separately, followed by univariate analyses if school type was significant (*p* = 0.05).

We validated mGLMs by plotting Dunn-Smyth residuals against fitted values and covariate school type, and verifying no patterns were present [75, 82]. We determined the significance of school type using LRTs and by bootstrapping probability integral transform (PIT) residuals using 10,000 resampling iterations [83]. If school type was significant (*p* < 0.05), we ran univariate analyses on individual taxa. We adjusted univariate *p* values to correct for multiple testing using a step-down resampling algorithm [82], but otherwise our statistical approach remained unchanged from the multivariate parent models. We could not include school as a blocking variable to account for non-independence of drawings from the same school since *mvabund* requires an equal number of samples in each level of the blocking variable. Instead, we ran a sensitivity analysis on a random sample of 9 drawings from each school using the same statistical approach, since 9 was the smallest number of drawings collected from a single school.

**2.4.3. Taxonomic resolution.** We used chi-square tests of independence to test for differences in the proportion of animal and plant terms identifiable to species level between categories, and for differences in the proportion of animal and plant terms identifiable to species level within each category between school types. We excluded terms in the 'General' categories for both animals and plants from these analyses since these terms, by definition, are not identifiable to species, and, for plants, no terms in the 'General' category had taxonomic meaning (S2 Table).

# 3. Results

## 3.1. Differences between taxa

**3.1.1. Animals.** The most frequently represented animals in the children's drawings were mammals (represented in 80.5% of all drawings) and birds (represented in 68.6% of all drawings; S3 Table; Fig 1a). Just over half of drawings contained an insect (54.6%) or another kind of invertebrate (51.4%), and only 15.7% of drawings contained a reptile or amphibian. Around 1 in 3 drawings contained no invertebrates (32.4%), and 2.7% of drawings used only 'General' terms for animals in the labels or caption (S3 Table; Fig 1a). In total, 123 different terms were used for animals, and the most 'speciose' categories were 'Insects' (27 terms), 'Garden Birds' (23 terms), 'Other Birds' (22 terms) and 'Wild Mammals' (19 terms) (S2 Table). The most commonly used animal term was 'Birds', which appeared in 48.6% of drawings, followed by 'Squirrel' (29.9%) and 'Ant' (28.9%) (Fig 2a).

The percentage of animal terms identifiable to each taxonomic level was as follows: species: 43.9%, genus: 8.9%, family: 15.4%, order: 19.5%, class: 8.9%, phylum: 1.6%, kingdom: 1.6% (Fig 3a). The proportion of animal terms identifiable to species level was significantly different between categories ($\chi^2$ = 13.145, *df* = 6, *p* = 0.04079), with terms in the categories 'Domestic Mammals', 'Garden Birds', 'Wild Mammals' and 'Other Birds' being the most identifiable to species level, at 90%, 69.6%, 57.9% and 54.5% of terms respectively (Fig 3a). In contrast, just 18.5% of terms in 'Insects' and 14.3% of terms in 'Herpetofauna' were identifiable to species level, while no terms in the category 'Other Invertebrates' were identifiable to species level (Fig 3a).

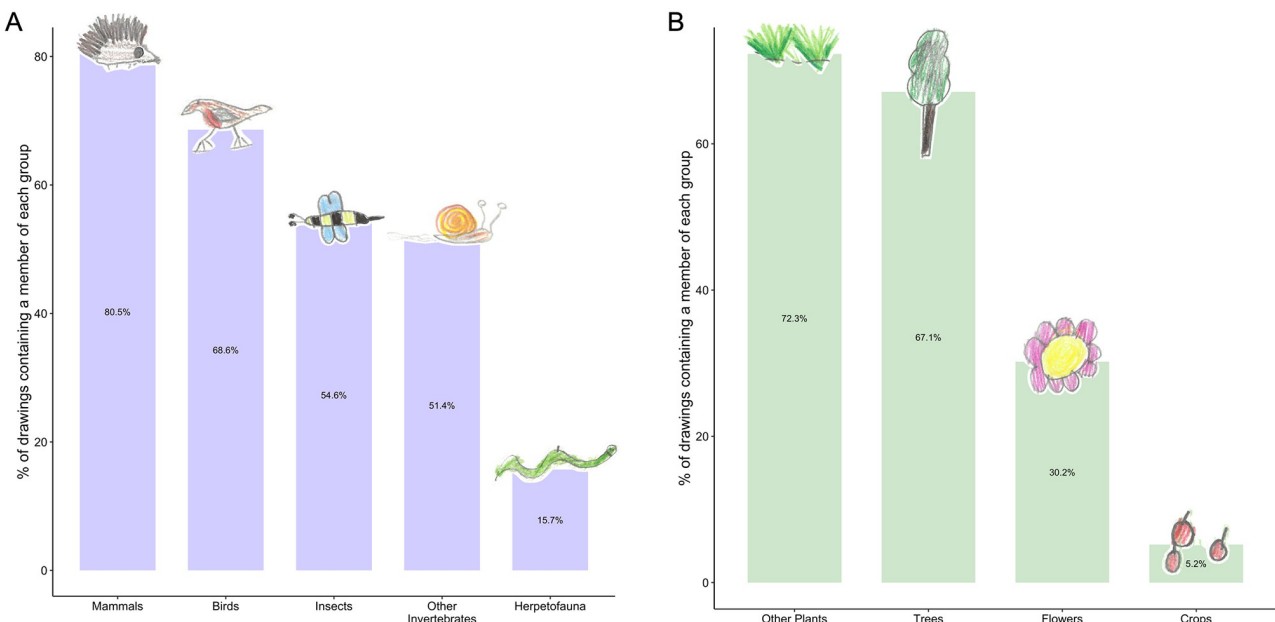

**Fig 1. Common animals and plants.** Histogram showing the percentage of drawings containing at least one representative of each group for A: animals and B: plants. These data represent the combined totals of both specific (e.g., 'Robin', 'Daisy') and general (e.g., 'Bird', 'Flower') representations.

**3.1.2. Plants.** 91.3% of drawings contained at least one kind of plant (S3 Table; Fig 1b), despite no explicit mention of plants in the instructions given to the children (S1 Appendix). 72.3% of drawings contained a plant that was not a tree, flower or crop, mostly due to frequent use of the terms 'grass' (59.1%) and 'bush' (23.2%) (Fig 1b). Taking account of both specific (e.g., 'Apple tree', 'Rose') and general (e.g., 'Trees', 'Flowers') representations, trees featured in 67.1% of drawings and flowers in 30.2% of drawings. 8.7% of drawings contained no plants,

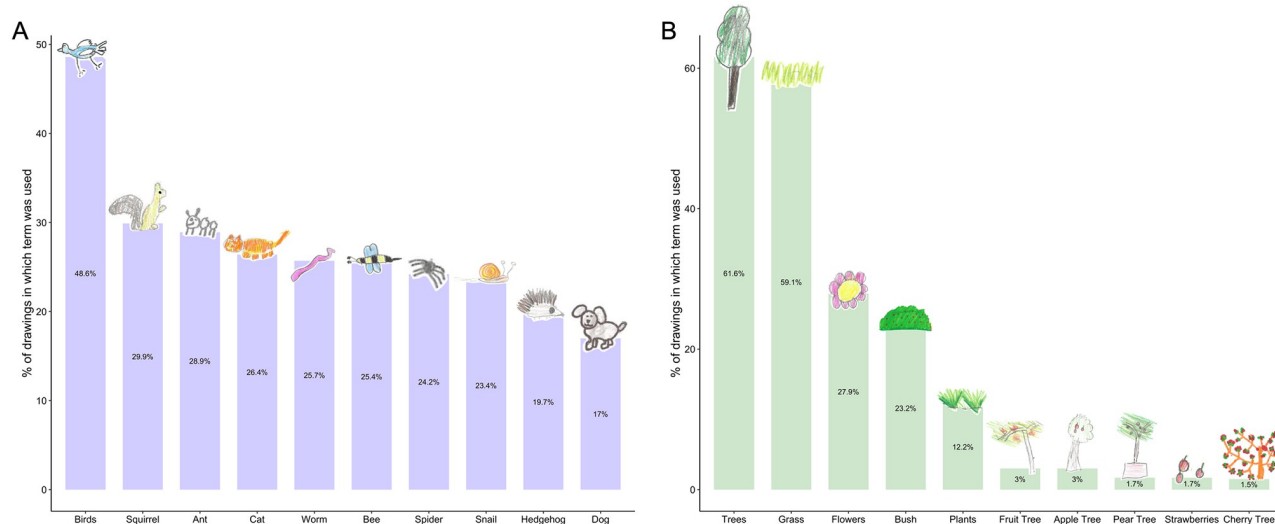

**Fig 2. Most commonly used terms.** Histogram showing the percentage of drawings containing each of the ten most commonly used terms for A: animals and B: plants.

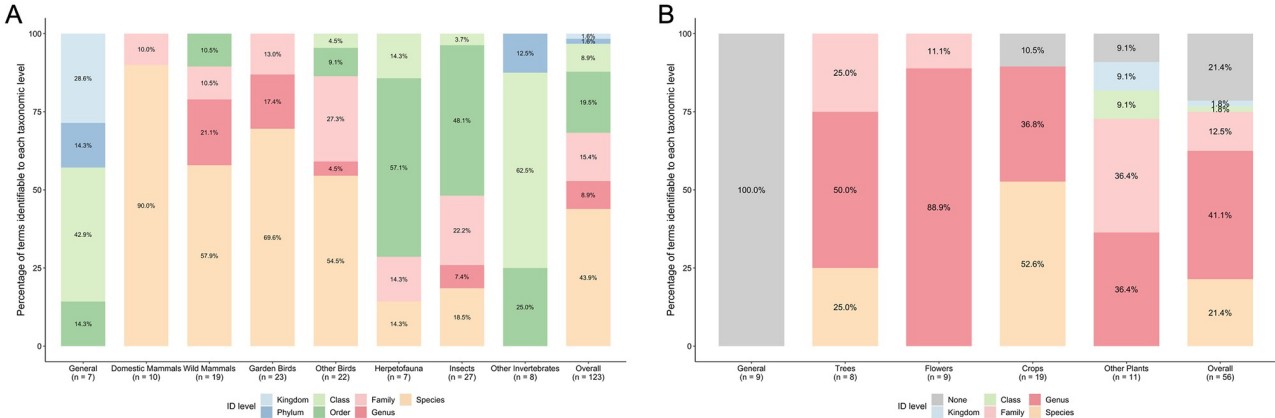

**Fig 3. Taxonomic resolution of terms.** Bar graphs showing the percentage of terms used in the labels and captions of children's drawings identifiable to each taxonomic level. Separate bars show percentages for each category, and far right bars show overall percentages with all categories combined. Bars are coloured by taxonomic level of identification. A: Animal terms ($n$ = 123). B: Plant terms ($n$ = 56).

and 26.4% of drawings used only 'General' terms for plants in the labels or captions (S3 Table; Fig 1b). 56 different terms were used for plants, and the most 'speciose' category was 'Crops' (19 terms) (S2 Table). The most commonly used plant term was 'Trees', which appeared in 61.6% of drawings, followed by 'Grass' (59.1%) and 'Flowers' (27.9%) (Fig 2b).

The percentage of plant terms identifiable to each taxonomic level was as follows: species: 21.4%, genus: 41.1%, family: 12.5%, order: 0%, class: 1.8%, phylum: 0%, kingdom: 1.8%; 21.4% of terms did not have taxonomic meaning (e.g., 'bush', 'flower', 'tree') (Fig 3b). The proportion of plant terms identifiable to species level was significantly different between categories ($\chi^2$ = 8.6877, $df$ = 3, $p$ = 0.03374), with terms in the categories 'Crops' and 'Trees' being the most identifiable to species level, at 52.6% and 25% of terms respectively (Fig 3b). No terms in 'Flowers' or 'Other Plants' were identifiable to species level (Fig 3b).

## 3.2. Differences between school types

**3.2.1. Species richness.** There was no effect of school type on animal species richness (LRT = 1.8706, $p$ = 0.3925; Fig 4a), but plant species richness differed between school types (LRT = 7.2149, $p$ = 0.02712; Fig 4b), with state school drawings having significantly higher plant species richness than those from private schools, although there was no difference between those from state and academy schools or between those from academy and private schools (S4 Table). There was no effect of buffer greenness on animal species richness (LRT = 0.14475, $p$ = 0.7036) or plant species richness (LRT = 1.6772, $p$ = 0.19530).

**3.2.2. Community composition.** Animal community composition was significantly different between school types (LRT = 53.19, $p$ < 0.0001; Fig 5a), with post-hoc tests showing this differed between all three pairs of school types (S4 Table). Univariate analyses indicated that the categories 'Garden Birds' (LRT = 10.091, $p$ = 0.0446) and 'Other Invertebrates' (LRT = 22.660, $p$ = 0.0001) were the primary drivers of these differences (Fig 5b), with a greater number of terms for garden birds listed in drawings from private schools than from other school types, and a greater number of terms for invertebrates (other than insects) listed in drawings from state schools than from other school types. There was a significant effect of buffer greenness on animal community composition (LRT = 15.91, $p$ = 0.047), although univariate analyses indicated that this was not driven by any group of animals in particular. However, neither the significant differences between school types, nor the effect of buffer

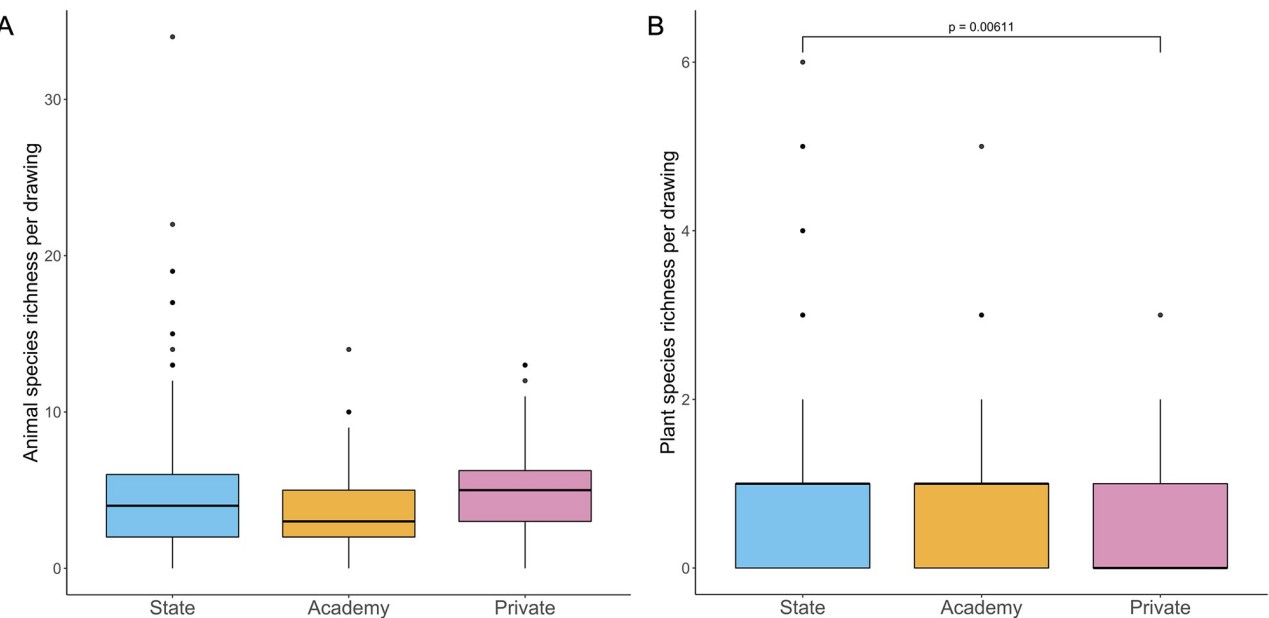

**Fig 4. Species richness.** Boxplots showing species richness of children's drawings (*n* = 401) by school type. Black lines indicate the median values. Coloured boxes show the interquartile ranges (IQR). Whiskers extend to the largest and smallest values no further than 1.5*IQR. A: Animal species richness by school type. B: Plant species richness by school type. Bracket labelled with adjusted *p* value shows a significant difference between state and private schools after post-hoc analyses.

greenness, were replicated in our sensitivity analysis (School type: LRT = 31.296, *p* = 0.178; Buffer greenness: LRT = 4.332, *p* = 0.765).

Plant community composition did not differ between school types (LRT = 13.45, *p* = 0.119 [sensitivity analysis: LRT = 15.305, *p* = 0.162]; Fig 6a and 6b), and there was no effect of buffer greenness (LRT = 5.614, *p* = 0.262 [sensitivity analysis: LRT = 3.284, *p* = 0.498]).

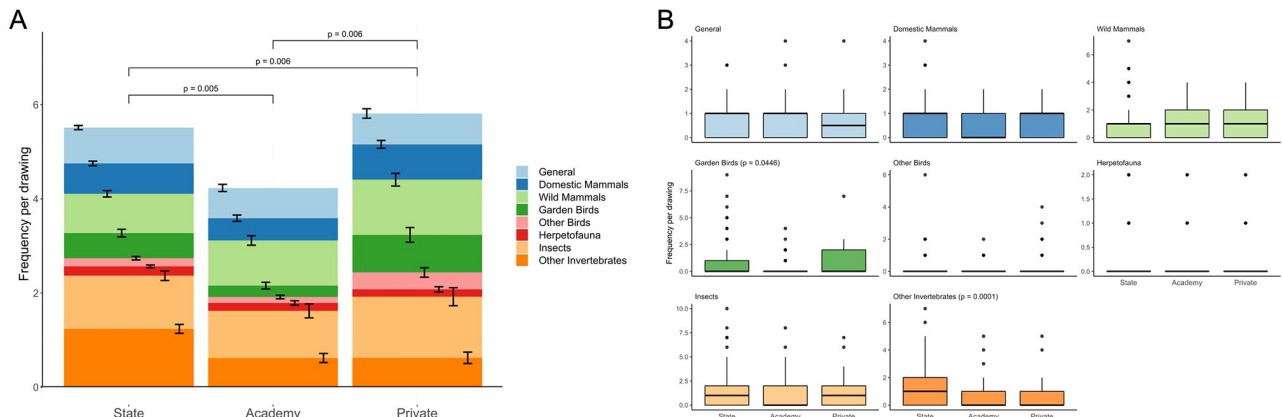

**Fig 5. Animal community composition.** Animal community composition of children's drawings (*n* = 401) by school type. A: Stacked bar graph showing category-level community composition with separate bars for each school type and coloured by category. Brackets labelled with *p* values show significant differences between pairs of school types. B: Boxplots showing category-level community composition by school type separated by category. Black lines indicate the median values. Coloured boxes show the interquartile ranges (IQR). Whiskers extend to the largest and smallest values no further than 1.5*IQR. *P* values are provided for the categories found to be primary drivers of differences following univariate analyses.

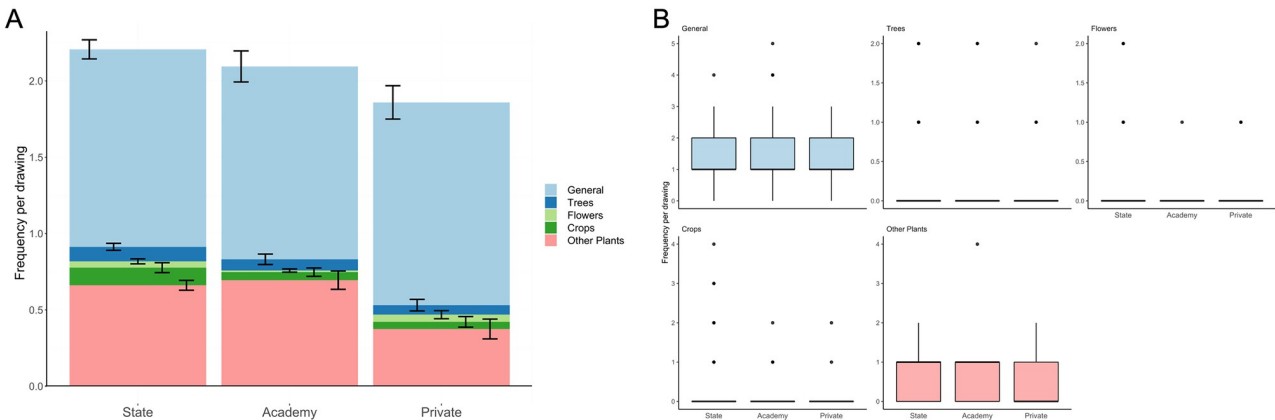

**Fig 6. Plant community composition.** Plant community composition of children's drawings (*n* = 401) by school type. A: Stacked bar graph showing category-level community composition with separate bars for each school type and coloured by category. B: Boxplots showing category-level community composition by school type separated by category. Black lines indicate the median values. Coloured boxes show the interquartile ranges (IQR). Whiskers extend to the largest and smallest values no further than 1.5*IQR.

**3.2.3. Taxonomic resolution.** Differences in the level of taxonomic resolution between animal categories were consistent across school types, with no significant differences within each category between school types (S5 Table). Differences in the level of taxonomic resolution between plant categories were also consistent across school types, with no significant differences within each category between school types (S6 Table).

## 4. Discussion

A wide range of animals and plants were represented in the children's drawings, with the majority (91.3%) containing plants, despite children not explicitly being asked to draw them. The most well represented group of animals was mammals, appearing in 80.5% of drawings, and the least well represented was herpetofauna, appearing in only 15.7% of drawings. Invertebrates appeared in just over half of drawings, with 54.6% containing at least one insect and 51.4% containing at least one invertebrate other than an insect. Just under half (43.9%) of all animal terms were identifiable to species. However, while most mammal (69%) and bird (62.2%) terms were identifiable to species, the majority of invertebrates and herpetofauna were not: just 18.5% of insects and 14.3% of herpetofauna were identifiable to species, while no invertebrates other than insects were identifiable to species. 21.4% of plant terms were identifiable to species, with all of these being crop plants or trees. Overall, we found no effect, or a limited effect, of school type on animal and plant species richness, community composition or taxonomic resolution of terms. Buffer greenness had no effect on species richness of drawings or plant community composition, but there was an effect on animal community composition, although this was not driven by any group of animals in particular. Collectively, these trends suggest that differences in ecological awareness between taxonomic groups amongst young children in England are fairly consistent and likely to reflect biases in portrayals in wider culture or innate biases, rather than differences in education. The 401 drawings used in this study were sourced from just 12 primary schools across England, and so represent a small sample compared to the numbers of children and primary schools in the country. The lack of differences we recorded between school types should therefore be treated with caution, as it is possible that a larger sample across a wider area could produce differing results. However, our findings of awareness across taxa are likely to be robust, since our analyses were conducted

across the whole sample and so are unlikely to be heavily influenced by individual school effects.

The greater representation of, and higher taxonomic resolution afforded to, mammals and birds over invertebrates and herpetofauna is consistent with discrepancies found in children's drawings of other schools in the UK [52] and the tropics [53], nature documentaries [84] and flagship species used by NGOs for fundraising campaigns [85], and mirrors preferences for species that are larger, more colourful, have forward-facing eyes and are phylogenetically or physically similar to humans [85, 86]. These same taxa are also more likely to be ranked as conservation priorities and attract monetary donations, trends which are mirrored in children as well as adults [61]. Generally, species with which people are less familiar and perceive as dangerous, such as reptiles and invertebrates, are perceived more negatively [60], potentially in part explaining their relative rarity in the children's drawings in our study. Indeed, conservation research over the last three decades has given greater focus to vertebrate groups than to invertebrate and plant groups [87]. The higher representation of mammals and birds over invertebrates and herpetofauna within children's drawings is therefore consistent with current understanding of people's perceptions of wildlife. It is difficult to unpick the directionality of this relationship—preferences for certain taxa could both influence and be determined by representations in wider culture [86]—but our findings complement another recent UK-based school study [52] in demonstrating that this skew in awareness of biodiversity towards mammals and birds, and away from invertebrates and herpetofauna, is already apparent in children as young as 7–11 years old.

The high rate of plant representation in drawings, despite children not explicitly being asked to draw plants, is consistent with previous research on children's perceptions of rainforests, which found that children drew a wide variety of vegetative components and non-animal habitat features when asked to draw their ideal rainforest while on a museum visit in the UK [53]. The generally low level of taxonomic resolution for plants found here, and the high inclusion of general, common terms, such as 'grass' and 'trees', over specific ones also mirrors findings from these rainforest drawings, which included a similarly high frequency of trees but little specificity beyond this [53]. This may also reflect the prominence of Cultivars within botanical taxonomy—a formal category in the International Code of Nomenclature for Cultivated Plants [88], which groups plants within the same genus that share defined characteristics [89], and may mean that genus level terms are more familiar. The lack of specificity for plants that we recorded reflects current understanding of people's perceptions [90], with public identification of plants being less accurate than that of animals [60, 91].

We found no effect of school type on animal species richness, but drawings from children at state schools had significantly higher plant species richness than those from children at private schools. There were also effects of school type and buffer greenness on animal community composition but not on plant community composition. However, the effects of school type and buffer greenness on animal community composition were not replicated in our sensitivity analysis, so were likely to be driven by uneven sample sizes across school types. There were no differences between the taxonomic resolution of terms used between school types. Collectively, these results suggest that children's awareness of local biodiversity is little influenced by school type, but instead by factors external to their education, such as exposure to wildlife in a home setting or representations of nature in wider culture (e.g., television programmes, social media or nature-inspired consumer products), despite the differences in budgets and curriculum-based requirements between state-funded and private education settings. The limited effect of residential greenness on ecological awareness suggests that passive exposure to greener surroundings is not sufficient to produce an ecologically accurate awareness of local biodiversity,

indicating that active engagement with overlooked taxa, such as invertebrates and herpeto-fauna, may be necessary to address the disparate awareness across taxa recorded here. Encouragingly, our findings also suggest that neither wealth nor residential greenness are primary determinants of ecological awareness, and that similar approaches to improve children's connection to and engagement with nature should be made across school types and degrees of urbanisation.

Our findings that children's drawings mirror taxonomic biases found in conservation literature [87], nature documentaries [84] and marketing materials used by NGOs [85] suggest either that these biases are innate or that representations in wider culture or family settings have already coloured young children's perceptions of biodiversity across school types by the time they enter secondary education. This matters because how species are perceived can have indirect effects on their risk of extinction [86]: the prominence of a species in wider society (societal salience) can make it more or less at risk of societal extinction (the loss of collective memory of a species, through the loss of attention, knowledge and representations associated with it from cultures and societies). This, in turn, can have knock-on effects on its risk of biological extinction, through difficulties in attracting conservation funding for relevant habitats or for the species directly. Therefore, the fact that young children's impressions of local biodiversity is skewed away from invertebrate, plant and herpetofauna groups, despite their vital roles in ecosystem functioning and high levels of biodiversity [92, 93], has the potential to place organisms within these overlooked groups at higher risk of societal extinction, making it harder to secure conservation funding to mitigate and avoid future species losses.

Interpretation of our findings should also take into account the following caveats. Firstly, using children's drawings to quantify awareness comes with potential disadvantages, such as children feeling limited to including features they feel able to draw. However, for children aged 7–11, asking them to write the names of animals and plants could be more limiting, due to requiring knowledge of spelling and writing, while children of this age generally enjoy drawing. Secondly, the instructions provided did not explicitly ask about plants. However, since 91.3% of drawings contained at least one plant, we felt plants merited analysis, albeit in light of this caveat. Finally, our sample was not balanced across the different school types or greenness of surrounding area, which is why we have used broad-brush tests of difference rather than more complex mixed-effects models, making our results robust to influences of particular schools.

We recommend that the skewed perception of biodiversity we have recorded among young children in England be addressed through targeted adjustments to national curricula, starting in early years teaching and continuing through to the age of 18, and that these adjustments should be applied across state-funded and private education settings. The current focus on improving climate and carbon literacy within the UK, through the introduction of sustainability leads at all nurseries, schools and higher education institutions in England [94], is welcome and creates an ideal climate in which to better integrate biodiversity literacy within school-based education. Indeed, the introduction of the new natural history GCSE makes a start on this [94], but this is optional and neglects biodiversity-focused education before the age of 14, and most other additions are heavily weighted towards carbon and climate, with relatively little attention currently paid to biodiversity conservation. This mirrors the relative media attention afforded to these dual issues, with climate change garnering up to eight times greater coverage across the USA, Canada and the UK than biodiversity loss [95]. Given that policy, scientific research and public opinion are part of a complex positive feedback mechanism [62], it is vital that tomorrow's young adults are also equipped with the knowledge, skills and awareness to understand and conserve biodiversity.

## Supporting information

**S1 Fig. School locations.** Locations of primary schools across England from which drawings were collected, coloured by school type. Each dot represents one school ($n = 12$). The private school in the main cluster of points has been manually moved 0.1 degrees west, so that all points are visible.
(DOCX)

**S1 Table. Drawings by school type and child age.** Table showing the distribution of children's drawings ($n = 401$) across ages and schools ($n = 12$), including school location and type.
(DOCX)

**S2 Table. Terms and categories.** Table showing all terms used in the labels and captions of children's drawings, detailing assigned categories, whether or not a term's presence was counted towards the species richness of a drawing, and the count and ID level for each term. Despite technically being the same species, 'Pony' and 'Horse' are both counted towards species richness because, to a child, these are different animals. 'Mushroom' is included in the category 'Other Plants' since it is the only fungal representative, and it appeared in just three drawings.
(DOCX)

**S3 Table. Relative popularity of animal and plant groups.** Table showing the numbers and proportions of children's drawings ($n = 401$) that contained at least one representative of each group (see S2 Table for a full list of terms and categories).
(DOCX)

**S4 Table. Post-hoc analyses results.** Table showing results of post-hoc analyses following a significant effect of school type in the parent model. For plant species richness, $p$ values are the results of Tukey all-pair comparisons, adjusted via the Bonferroni correction, following a significant effect of school type in the GLMM. State school drawings had higher species richness than private school drawings. For animal community composition, $p$ values are the results of pairwise comparisons via an analysis of deviance for the mGLM, adjusted for multiple comparisons via a free step-down resampling procedure. State school drawings contained more terms for invertebrates other than insects, and private school drawings contained more terms for garden birds. Significant $p$ values are shown in bold.
(DOCX)

**S5 Table. Results of chi-square tests on taxonomic resolution of animal terms.** Results of chi-square tests of independence on differences in taxonomic resolution of animal terms in each category by school type.
(DOCX)

**S6 Table. Results of chi-square tests on taxonomic resolution of plant terms.** Results of chi-square tests of independence on differences in taxonomic resolution of plant terms in each category by school type.
(DOCX)

**S1 Appendix. Drawing worksheet.** Worksheet with instructions given to primary-school children. Font size and box dimensions have been reduced in order to fit within document margins.
(DOCX)

**S2 Appendix. Participant Information and written consent.** Information provided to parents or guardians of children before the study.
(DOCX)

## Acknowledgments

We thank the Public Engagement team at the University Museum of Zoology, Cambridge, and all participating schools, teachers and students for their support with this project.

## Author Contributions

**Conceptualization:** Kate Howlett.

**Data curation:** Kate Howlett.

**Formal analysis:** Kate Howlett.

**Investigation:** Kate Howlett.

**Methodology:** Kate Howlett.

**Project administration:** Kate Howlett.

**Supervision:** Edgar C. Turner.

**Writing – original draft:** Kate Howlett.

**Writing – review & editing:** Edgar C. Turner.

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
