## [Decision Letter · Decision Letter 0]

22 Nov 2022

PONE-D-22-22596What can drawings tell us about children’s perceptions of nature?PLOS ONE

Dear Dr. Howlett,

Thank you for submitting your manuscript to PLOS ONE. After careful consideration, we feel that it has merit but does not fully meet PLOS ONE’s publication criteria as it currently stands. Therefore, we invite you to submit a revised version of the manuscript that addresses the points raised during the review process.

We look forward to receiving your revised manuscript.

Kind regards,

Daniel de Paiva Silva, Ph.D.

Academic Editor

PLOS ONE

Journal Requirements:

“K.H. is funded by the Natural Environment Research Council (grant number NE/L002507/1).”

“K.H. is funded by the Natural Environment Research Council (grant number NE/L002507/1): https://www.ukri.org/councils/nerc/. The funder had no role in study design, data collection and analysis, decision to publish, or preparation of the manuscript.”

Additional Editor Comments (if provided):

Dear Dr. Howlett,

After this first review round, the reviewers believe your manuscript may be accepted for publication in PLoS One after a thourough revision. Please consider all the issues raised by both reviewers and provide a rebuttal letter until February 21st 2023. Do not hesitate to write me in case you have any doubts. Also, do not hesitate to resubmit earlier in case you are able to.

Sincerely,

Daniel Silva

Reviewers' comments:

Reviewer's Responses to Questions

**Comments to the Author**

1. Is the manuscript technically sound, and do the data support the conclusions?

Reviewer #1: Yes

Reviewer #2: Partly

2. Has the statistical analysis been performed appropriately and rigorously? 

Reviewer #1: I Don't Know

Reviewer #2: Yes

3. Have the authors made all data underlying the findings in their manuscript fully available?

Reviewer #1: Yes

Reviewer #2: Yes

4. Is the manuscript presented in an intelligible fashion and written in standard English?

Reviewer #1: Yes

Reviewer #2: Yes

5. Review Comments to the Author

Reviewer #1: Research is very interesting and analyzing drawings is a challenging task. so congratulations on the methodology. It was interesting to know that the students' perception of biodiversity does not depend on the type of school administration, this reveals that the country's educational system is egalitarian. I was curious to know if your result is compatible with other countries or the international context.

Reviewer #2: This research article reports the findings of a study assessing primary school children’s perception of nature via a drawing exercise, whereby participants were asked to draw and label the animals found within either their garden or local green space. This concise article reporting the findings from a simple study is largely written in a clear manner and the analyses performed are appropriate for the obtained data and research questions of interest. However, the article does require stronger framing, theoretical grounding and embedding of relevant existing studies/literature. The rationale for the study needs to be more transparent. Furthermore, the article is missing a critical discussion of the utility of drawing exercises for such research.

1) Currently, the intro and discussion do not acknowledge and incorporate the findings of previous drawing studies that have investigated children’s perceptions of nature. It is not clear why this omission has been made and it needs to be addressed so that your research is grounded within the body of existing studies. I’ve provided several example studies below – in particular, Montgomery et al. (2022) seems particularly pertinent wherein biodiversity perception was assessed by children drawing what they thought was in their school grounds. Children initially perceived few organisms within easily visible taxa, and perceived more vertebrates compared to invertebrate species. After, children were more aware of taxa, resulting in a more reflective biodiversity perception.

Aaron, R.F. and Witt, P.A., 2011. Urban students' definitions and perceptions of nature. Children Youth and Environments, 21(2), pp.145-167.

Drissner, J.R., Haase, H.M., Wittig, S. and Hille, K., 2014. Short-term environmental education: long-term effectiveness? Journal of Biological Education, 48(1), pp.9-15.

Montgomery, L.N., Gange, A.C., Watling, D. and Harvey, D.J., 2022. Children’s perception of biodiversity in their school grounds and its influence on their wellbeing and resilience. Journal of Adventure Education and Outdoor Learning, pp.1-15. https://doi.org/10.1080/14729679.2022.2100801

Morón-Monge, H., Hamed, S. and Moron Monge, M.D.C., 2021. How Do Children Perceive the Biodiversity of Their nearby Environment: An Analysis of Drawings. Sustainability, 13(6), p.3036.

Moula, Z., Walshe, N., & Lee, E. (2021). Making nature explicit in children’s drawings of wellbeing and happy spaces. Child Indicators Research, 14(4), 1653–1675.

Prokop, P., Prokop, M. and Tunnicliffe, S.D., 2008. Effects of keeping animals as pets on children's concepts of vertebrates and invertebrates. International Journal of Science Education, 30(4), pp.431-449.

2) The study doesn’t critically discuss the utility of drawings for understanding children’s perception of nature. Missing a critique of drawing analysis – what are the strengths and what are the weaknesses? See my specific comments in the methods/discussion section below.

3) The rationale for this study is not 100% transparent at the moment. It seems to be the case that you propose understanding children’s perception/knowledge of local species has important consequences for understanding and facilitating nature connection and pro- environmental/conservation behaviours? However, a number of studies/perspective pieces exist which question the importance of ecological knowledge (including species identification) in nature connection/behaviours (e.g. Lumber et al. 2017, below). The paragraph startling Line 72 in the intro is very knowledge focused and it isn’t clear why.

Lumber, R., Richardson, M. and Sheffield, D., 2017. Beyond knowing nature: Contact, emotion, compassion, meaning, and beauty are pathways to nature connection. PLoS one, 12(5), p.e0177186.

Please find below some more specific feedback.

INTRODUCTION

• Line 63: the drivers of the extinction of experience are more nuanced than presented here.

• Line 82: Recommend reading and integrating key associated points from the following papers:

Gaston, K.J. and Soga, M., 2020. Extinction of experience: The need to be more specific. People and Nature, 2(3), pp.575-581.

Soga, M., Gaston, K.J., Yamaura, Y., Kurisu, K. and Hanaki, K., 2016. Both direct and vicarious experiences of nature affect children’s willingness to conserve biodiversity. International journal of environmental research and public health, 13(6), p.529.

Soga, M. and Gaston, K.J., 2020. The ecology of human–nature interactions. Proceedings of the Royal Society B, 287(1918), p.20191882.

• Line 86: More residential green space or more accessible residential green space? Needs clarifying.

• Near where you first introduce the concept of nature connection, a working definition should be provided.

• Paragraph starting Line 96: Make the distinction between pro-environmental behaviour and pro nature conservation behaviour, as the two are different, but see to be used interchangeable in this parargaph – see: Richardson, M., Passmore, H.A., Barbett, L., Lumber, R., Thomas, R. and Hunt, A., 2020. The green care code: How nature connectedness and simple activities help explain pro‐nature conservation behaviours. People and Nature, 2(3), pp.821-839.

• Sentence starting on Line 97: avoid sweeping statements/contradictions. For example, although there is less evidence for the latter, both direct and vicarious experiences with nature have been found to enhance nature connection and associated behaviours; e.g. Soga, M., Gaston, K.J., Yamaura, Y., Kurisu, K. and Hanaki, K., 2016. Both direct and vicarious experiences of nature affect children’s willingness to conserve biodiversity. International journal of environmental research and public health, 13(6), p.529; and Hughes, J., Richardson, M. and Lumber, R., 2018. Evaluating connection to nature and the relationship with conservation behaviour in children. Journal for Nature Conservation, 45, pp.11-19.

• Citation 38 is for a study focused on adults not children?

• Citation missing to support sentence starting on Line 112.

• Introduction is missing a definition of ecological knowledge and its connection to nature connection and related dimensions such as pro environmental/conservation behaviours.

• Clear aim and questions, but prior to conducting the study did you have any a priori hypotheses?

METHODS

• It seems as if the framing of the drawing task could potentially lead to bias in the results. Some gardens or green spaces will be more biodiverse than others. Furthermore, a child may choose to draw the animals in their garden which might be species poor but actually have higher ecological knowledge than displayed in the drawing? Furthermore, every individual has different drawing abilities – not being able to draw species (clearly) isn’t indicative of poor perception/knowledge? Also, some species are just simply more challenging to draw than others, which could influence inclusion or ability for the researchers to identify?

• In the instructions, were the children told what to include as a label? i.e. to provide the species name of each animal drawn or rather simply to just label their drawing?

• It must have been quite challenging trying to identify a number of the animals/plants drawn, and a number of assumptions consequently made?

• I do question the ecological rigour of some of the study variables. For example, it wasn’t really possible to quantify ‘species richness’; therefore, this variable heading seems a bit misleading – would ‘taxa richness’ be more appropriate? Furthermore, putting fungi in the ‘other plants’ category is also misleading.

• Info missing regarding what spatial maps/images you used to obtain buffer greenness.

• What was the rationale for analysing plants in this study when not explicitly asked for in the drawing task? If you had asked them to draw all of the animals and plants in their garden/local green space, there could have been some nice scope to assess the prevailing issue of plant blindness.

• It seems that domestic species were included in data analysis – what was the reason for this? Related to this, was any exploration undertaken to investigate frequency of exotic/non-native species vs. wild/native species drawn?

• Did you explore/identify any errors in the submitted drawings? E.g. exotic species in drawings that definitely wouldn’t have been seen in someone’s garden or local green space?

RESULTS

• Results presentation is a bit repetitive for Lines 388-391 – integrate stats outputs into s8 fig instead?

• Do you need Fig S9 if just showing non-sig results?

DISCUSSION

• There is currently limited critical discussion of the study protocol and/or potential confounding factors, and I think this does need to be considered and integrated here. For example, the issues/biases with the drawing exercise itself, the fact that the sample is self-selecting, and that the sampling design is not balanced across school type or greenness of surrounding area.

• L418-421: I don’t fully follow the rationale presented here.

• Citation 70 is currently in review? Typically can’t cite such publications until they become in press.

• L425-430: the discussion jumps here to focus on positive/negative sentiment towards species, and it’s not entirely clear why in the context of the study’s scope.

• Be careful throughout to make it clear which cited studies focus on adults and which on children – in some sentences/paragraphs, a combination of both are mixed together.

• Paragraph starting Line 422: I would be cautious here. Including a robin in a drawing doesn’t show preference for it.

• Where you compare you study and its findings to that of Snaddon et al., it needs to be clearly acknowledged that their study was quite different to yours – they asked primary-age children visiting the University Museum of Zoology in Cambridge to draw their ideal/rainforest.

• I don’t follow rationale presented on Lines 445-448.

• Lines 451-470: No citations. Can these conclusions have confidence given uneven sample size across school types, and the small number of schools included in the study?

• Line 492: But the new forthcoming Natural History GCSE also focuses on biodiversity conservation, ecology and natural history - not just sustainability and climate change.

• Last sentence (Line 498): But why is awareness and knowledge important of species biod – what are the mechanisms that link this to conservation? Final bit about climate change also seems tacked on.

6. PLOS authors have the option to publish the peer review history of their article (what does this mean?). If published, this will include your full peer review and any attached files.

Reviewer #1: No

Reviewer #2: No

---

## [Author Response · Author response to Decision Letter 0]

30 Jan 2023

Dear Dr Silva,

Thank you for coordinating the review process for our manuscript and to both reviewers for providing such helpful comments and suggestions, which have greatly improved the article.

We have responded to the reviewers’ comments point-by-point below.

Kind regards,

Kate Howlett

Response to Reviewers' Comments to the Author:

Reviewer #1: Research is very interesting and analyzing drawings is a challenging task. so congratulations on the methodology. It was interesting to know that the students' perception of biodiversity does not depend on the type of school administration, this reveals that the country's educational system is egalitarian. I was curious to know if your result is compatible with other countries or the international context.

Thank you for your positive comments on our paper. We have not come across results evaluating discrepancies in biodiversity awareness across educational systems in other countries, but children in France showed a similar bias towards large charismatic species over smaller local species (reference 61: Ballouard, Brischoux and Bonnet. 2011. ‘Children Prioritize Virtual Exotic Biodiversity over Local Biodiversity’. PLoS ONE.).

Reviewer #2: This research article reports the findings of a study assessing primary school children’s perception of nature via a drawing exercise, whereby participants were asked to draw and label the animals found within either their garden or local green space. This concise article reporting the findings from a simple study is largely written in a clear manner and the analyses performed are appropriate for the obtained data and research questions of interest. However, the article does require stronger framing, theoretical grounding and embedding of relevant existing studies/literature. The rationale for the study needs to be more transparent. Furthermore, the article is missing a critical discussion of the utility of drawing exercises for such research.

1) Currently, the intro and discussion do not acknowledge and incorporate the findings of previous drawing studies that have investigated children’s perceptions of nature. It is not clear why this omission has been made and it needs to be addressed so that your research is grounded within the body of existing studies. I’ve provided several example studies below – in particular, Montgomery et al. (2022) seems particularly pertinent wherein biodiversity perception was assessed by children drawing what they thought was in their school grounds. Children initially perceived few organisms within easily visible taxa, and perceived more vertebrates compared to invertebrate species. After, children were more aware of taxa, resulting in a more reflective biodiversity perception.

Aaron, R.F. and Witt, P.A., 2011. Urban students' definitions and perceptions of nature. Children Youth and Environments, 21(2), pp.145-167; Drissner, J.R., Haase, H.M., Wittig, S. and Hille, K., 2014. Short-term environmental education: long-term effectiveness? Journal of Biological Education, 48(1), pp.9-15; Montgomery, L.N., Gange, A.C., Watling, D. and Harvey, D.J., 2022. Children’s perception of biodiversity in their school grounds and its influence on their wellbeing and resilience. Journal of Adventure Education and Outdoor Learning, pp.1-15. https://doi.org/10.1080/14729679.2022.2100801; Morón-Monge, H., Hamed, S. and Moron Monge, M.D.C., 2021. How Do Children Perceive the Biodiversity of Their nearby Environment: An Analysis of Drawings. Sustainability, 13(6), p.3036; Moula, Z., Walshe, N., & Lee, E. (2021). Making nature explicit in children’s drawings of wellbeing and happy spaces. Child Indicators Research, 14(4), 1653–1675; Prokop, P., Prokop, M. and Tunnicliffe, S.D., 2008. Effects of keeping animals as pets on children's concepts of vertebrates and invertebrates. International Journal of Science Education, 30(4), pp.431-449.

Thank you for suggesting these papers. We have now incorporated all of these in lines 169-190. 

2) The study doesn’t critically discuss the utility of drawings for understanding children’s perception of nature. Missing a critique of drawing analysis – what are the strengths and what are the weaknesses? See my specific comments in the methods/discussion section below.

Thank you for suggesting we include critical discussion of the use of drawings. We have now added a paragraph to our discussion to address this in line with your helpful comments (lines 599-613).

3) The rationale for this study is not 100% transparent at the moment. It seems to be the case that you propose understanding children’s perception/knowledge of local species has important consequences for understanding and facilitating nature connection and pro- environmental/conservation behaviours? However, a number of studies/perspective pieces exist which question the importance of ecological knowledge (including species identification) in nature connection/behaviours (e.g. Lumber et al. 2017, below). The paragraph startling Line 72 in the intro is very knowledge focused and it isn’t clear why.

Lumber, R., Richardson, M. and Sheffield, D., 2017. Beyond knowing nature: Contact, emotion, compassion, meaning, and beauty are pathways to nature connection. PLoS one, 12(5), p.e0177186.

Thank you for raising this issue and for suggesting this paper. We have incorporated this paper and views, and have explained our rationale more explicitly in lines 180-190. While it is true that ecological knowledge may not be necessary for development of nature connection, we argue that familiarity with species affects development positive attitudes, which affects public opinion on which species require conservation.

Please find below some more specific feedback.

INTRODUCTION

• Line 63: the drivers of the extinction of experience are more nuanced than presented here.

Thank you for highlighting this. We have added more detail to this opening sentence (lines 64-65), along with a reference to your suggested Gaston and Soga 2020 paper, ‘Extinction of experience: The need to be more specific’.

• Line 82: Recommend reading and integrating key associated points from the following papers:

Gaston, K.J. and Soga, M., 2020. Extinction of experience: The need to be more specific. People and Nature, 2(3), pp.575-581; Soga, M., Gaston, K.J., Yamaura, Y., Kurisu, K. and Hanaki, K., 2016. Both direct and vicarious experiences of nature affect children’s willingness to conserve biodiversity. International journal of environmental research and public health, 13(6), p.529; Soga, M. and Gaston, K.J., 2020. The ecology of human–nature interactions. Proceedings of the Royal Society B, 287(1918), p.20191882.

Thank you for suggesting these three relevant papers. We have now added discussion of their relevant points in lines 88-103.

• Line 86: More residential green space or more accessible residential green space? Needs clarifying.

Now clarified in lines 106-108.

• Near where you first introduce the concept of nature connection, a working definition should be provided.

Thank you for suggesting this. We have now added this in lines 86-88.

• Paragraph starting Line 96: Make the distinction between pro-environmental behaviour and pro nature conservation behaviour, as the two are different, but see to be used interchangeable in this parargaph – see: Richardson, M., Passmore, H.A., Barbett, L., Lumber, R., Thomas, R. and Hunt, A., 2020. The green care code: How nature connectedness and simple activities help explain pro‐nature conservation behaviours. People and Nature, 2(3), pp.821-839.

Thank you for highlighting this distinction. We have now incorporated the suggested paper in line 123 and specified ‘pro-nature conservation’ (line 124) or ‘pro-environmental’ (line 127) behaviour where appropriate.

• Sentence starting on Line 97: avoid sweeping statements/contradictions. For example, although there is less evidence for the latter, both direct and vicarious experiences with nature have been found to enhance nature connection and associated behaviours; e.g. Soga, M., Gaston, K.J., Yamaura, Y., Kurisu, K. and Hanaki, K., 2016. Both direct and vicarious experiences of nature affect children’s willingness to conserve biodiversity. International journal of environmental research and public health, 13(6), p.529; and Hughes, J., Richardson, M. and Lumber, R., 2018. Evaluating connection to nature and the relationship with conservation behaviour in children. Journal for Nature Conservation, 45, pp.11-19.

Thank you for highlighting this. We have now made the evidence for vicarious experiences clear in lines 121-122.

• Citation 38 is for a study focused on adults not children?

Thank you for pointing this out. We have now made this clear in line 124.

• Citation missing to support sentence starting on Line 112.

Thank you for pointing out this omission. We have now added the relevant citation (now line 135).

• Introduction is missing a definition of ecological knowledge and its connection to nature connection and related dimensions such as pro environmental/conservation behaviours.

Thank you for highlighting this. We have now included working definitions of ecological knowledge (lines 83-84) and nature connection (lines 86-88), and have been more specific when referring to pro-nature conservation and pro-environmental behaviours (lines 124-128).

• Clear aim and questions, but prior to conducting the study did you have any a priori hypotheses?

We have added our a priori hypotheses in lines 206-210.

METHODS

• It seems as if the framing of the drawing task could potentially lead to bias in the results. Some gardens or green spaces will be more biodiverse than others. Furthermore, a child may choose to draw the animals in their garden which might be species poor but actually have higher ecological knowledge than displayed in the drawing? Furthermore, every individual has different drawing abilities – not being able to draw species (clearly) isn’t indicative of poor perception/knowledge? Also, some species are just simply more challenging to draw than others, which could influence inclusion or ability for the researchers to identify?

Thank you for raising these issues with our methodology. We have now added a paragraph in which we directly raise and discuss these in lines 599-613. We agree with all of the points raised. We argue that drawing is less restrictive than writing, since it does not require knowledge of spelling or writing, and children generally enjoy drawing, although of course this is still restricted. We agree that their ecological knowledge is not directly equivalent to asking them what animals they think live in a particular place, but we wanted to tie their thinking to a particular place rather than asking them to simply draw or list all the UK animals they could think of. This exercise could be seen by children as a test of knowledge rather than an exercise related to local wildlife, so they might be likely to start listing all animals they can think of – elephants, tigers etc.

• In the instructions, were the children told what to include as a label? i.e. to provide the species name of each animal drawn or rather simply to just label their drawing?

We did not ask the children to provide a species name of each animal because this was judged to be too challenging, and we felt might hinder which animals they included in their drawings. We simply asked them to label their drawings however they saw fit. This then enabled us to explore the precision with which they labelled animals and plants of different taxa. All the instructions given to the children are included in S3 Appendix and in lines 245-256, along with our rationale. 

• It must have been quite challenging trying to identify a number of the animals/plants drawn, and a number of assumptions consequently made?

Thank you for raising this. We did not record an animal or plant as different from another unless it was unambiguously different (see lines 279-286) (e.g., if multiple birds were drawn, ‘Bird’ was recorded as present, unless there were obvious diagnostic features present, such as a black body and yellow beak, or a red breast, in which case both ‘Robin’ and ‘Blackbird’ were recorded as present). We feel we have been as cautious and rigorous with this methodology as reasonably possible, and hope our explanations satisfy your concerns.

• I do question the ecological rigour of some of the study variables. For example, it wasn’t really possible to quantify ‘species richness’; therefore, this variable heading seems a bit misleading – would ‘taxa richness’ be more appropriate? Furthermore, putting fungi in the ‘other plants’ category is also misleading.

Thank you for raising these points, both of which we agree with. For this reason, we have tried to be as explicit as possible about the meaning of ‘species richness’ in the context of this paper (lines 289-292) and about the inclusion of ‘mushroom’ in ‘Other plants’ (line 306-307, and in the caption of S5 Table) so that it is not misleading for readers. With regards to ‘species richness’, we have opted for this term instead of ‘taxa richness’ on the basis of a reviewer’s comment on a previous, related paper that ‘species’ is understandable by a broader audience than ‘taxa’. We have tried to make this clearer, and that we do not mean this to be interpreted in its strictly ecological sense, in lines 290-292.

• Info missing regarding what spatial maps/images you used to obtain buffer greenness.

Thank you for pointing this out. We have added detail of the images used in line 320.

• What was the rationale for analysing plants in this study when not explicitly asked for in the drawing task? If you had asked them to draw all of the animals and plants in their garden/local green space, there could have been some nice scope to assess the prevailing issue of plant blindness.

Thank you for raising this point. We agree that it would have been a good idea to have explicitly asked the children to include plants. We felt that, since 91.3% of all drawings included at least one plant, this was still worth analysing, as it was clear that children were very aware of them and connected them with their local green spaces. We have now included this as an explicit caveat in lines 608-610, and we also point this out when we present the plant-based results in lines 409-410.

• It seems that domestic species were included in data analysis – what was the reason for this? Related to this, was any exploration undertaken to investigate frequency of exotic/non-native species vs. wild/native species drawn?

Thank you for flagging this point. Our rationale for separating out domestic mammals from wild mammals was that domestic mammals featured in such a high proportion of drawings and with such high diversity (i.e., a high number of different domestic mammals per drawing). If we had not made this distinction, interpretation of the results would be more challenging since we would not be able to tell whether the high representation of mammals was true across all mammals, or due more or less to either domestic or wild mammals. We feel that this classification adds interpretation and colour to our results. We have added this rationale to the methods in lines 299-303.

We did not investigate non-native versus native species since the vast majority of all animals and plants drawn were native. Indeed, the highest proportion of non-native species were domestic mammals (e.g., guinea pigs), so we did not feel that this question merited further investigation.

• Did you explore/identify any errors in the submitted drawings? E.g. exotic species in drawings that definitely wouldn’t have been seen in someone’s garden or local green space?

Thank you for asking this. We did not explore any errors in the submitted drawings for the same rationale as the previous comment. There was a very low number of these kind of ‘errors’ in the drawings, if any, so we did not feel this merited investigation.

RESULTS

• Results presentation is a bit repetitive for Lines 388-391 – integrate stats outputs into s8 fig instead?

Thank you for suggesting this. We have now incorporated all of this information into a table (S8 Table) to replace S8 Fig, given that the results are not significant.

• Do you need Fig S9 if just showing non-sig results?

Similar to the comment above, we have now incorporated all of this information into a table (S9 Table) to replace S9 Fig, given that the results are not significant.

DISCUSSION

• There is currently limited critical discussion of the study protocol and/or potential confounding factors, and I think this does need to be considered and integrated here. For example, the issues/biases with the drawing exercise itself, the fact that the sample is self-selecting, and that the sampling design is not balanced across school type or greenness of surrounding area.

Thank you for raising these points and for suggesting we expand our discussion of the protocol’s limitations. We have now added a paragraph doing this in lines 599-613. We argue that drawing is less restrictive than writing, and we have highlighted again the imbalance across school types. By analysing our data through simple tests of difference rather than more complex mixed-effects models, we hope that our results are robust to the idiosyncrasies of particular schools.

• L418-421: I don’t fully follow the rationale presented here.

Thank you for pointing out that this wasn’t clear. We have changed the wording here (now lines 511-514) to make our logic clearer. Our point is that all our analyses were conducted across the whole sample, rather than on smaller subsets of the data (e.g., one school type), so our results should be robust to idiosyncrasies of particular schools.

• Citation 70 is currently in review? Typically can’t cite such publications until they become in press.

Thank you for highlighting this. This paper has now been accepted and is in press, so we have updated the citation accordingly (now citation 84).

• L425-430: the discussion jumps here to focus on positive/negative sentiment towards species, and it’s not entirely clear why in the context of the study’s scope.

Thank you for highlighting this. We have reworded this paragraph (now lines 515-537) to make our argument clearer. This paragraph is making the point that the species we see most used for conservation marketing campaigns are the same ones that people perceive positively and the same ones that people list as conservation priorities and have the greatest awareness of. While it might not seem logical that positive/negative sentiment is relevant to what wildlife people are most aware of, it clearly is, and this in turn affects which species attract conservation funding, so it also matters, whether or not it is logical.

• Be careful throughout to make it clear which cited studies focus on adults and which on children – in some sentences/paragraphs, a combination of both are mixed together.

Thank you for pointing this out. We have tried to adhere to this throughout, as suggested.

• Paragraph starting Line 422: I would be cautious here. Including a robin in a drawing doesn’t show preference for it.

Thank you for highlighting this. As above, we have now reworded this paragraph to make our argument tighter (now lines 515-537). This paragraph is attempting to pull together the apparently disparate threads of positive/negative sentiment, conservation attention and ecological awareness (see comment two above). While drawing a robin doesn’t necessarily show a preference for it, there are nonetheless commonalities between the animals we see most represented in the media, most often listed as conservation priorities and most preferred. This is why we need to ensure a more representative portrayal of the natural world in wider media—the causality of the relationship between species preference and awareness is unknown, but the two are certainly linked.

• Where you compare you study and its findings to that of Snaddon et al., it needs to be clearly acknowledged that their study was quite different to yours – they asked primary-age children visiting the University Museum of Zoology in Cambridge to draw their ideal/rainforest.

Thank you for this suggestion. We have now made this more explicit (e.g., lines 541 and 544).

• I don’t follow rationale presented on Lines 445-448.

Thank you for highlighting this. We have clarified our thinking in line 553. Our point here is that people are often familiar with plants in this broad-category sense because of the prominence of cultivars and therefore how they are referred to, and so this might explain why so few of the plants were identifiable to species level.

• Lines 451-470: No citations. Can these conclusions have confidence given uneven sample size across school types, and the small number of schools included in the study?

Thank you for raising this. We have adjusted the wording in lines 570-572 to make it clear that we are referring to biases between taxa and not school types, which made our interpretation seem more wider-reaching than we had intended. This paragraph (now lines 556-582) is entirely description and interpretation of our own results, so we are not sure where a citation would be relevant. 

• Line 492: But the new forthcoming Natural History GCSE also focuses on biodiversity conservation, ecology and natural history - not just sustainability and climate change.

Thank you for pointing this out. We have removed our reference to the new natural history GCSE from the sentence in lines 617-621 and made explicit discussion of this in 621-623. While this new GCSE is welcome, it remains optional and does not tackle the problem of a lack of biodiversity-focused education before the age 14.

• Last sentence (Line 498): But why is awareness and knowledge important of species biod – what are the mechanisms that link this to conservation? Final bit about climate change also seems tacked on.

Thank you for raising this. We have now reworded our final sentence to mirror the focus of our paper better (lines 627-629). In this final sentence, we are making the point that we should be prioritising education on biodiversity and ecology, in addition to education around climate change, given that public awareness influences policy and scientific research in a complex network of effects that we don’t yet understand (see reference in line 628). We have discussed in this paragraph the disparate attention given to climate change versus biodiversity, which is why this is referred to in the final sentence.

---

## [Decision Letter · Decision Letter 1]

29 May 2023

PONE-D-22-22596R1What can drawings tell us about children’s perceptions of nature?PLOS ONE

Dear Dr. Howlett,

Thank you for submitting your manuscript to PLOS ONE. After careful consideration, we feel that it has merit but does not fully meet PLOS ONE’s publication criteria as it currently stands. Therefore, we invite you to submit a revised version of the manuscript that addresses the points raised during the review process.

We look forward to receiving your revised manuscript.

Kind regards,

Daniel de Paiva Silva, Ph.D.

Academic Editor

PLOS ONE

Journal Requirements:

Additional Editor Comments:

Dear Dr. Howlett,

Thank you for your patience and support regarding the indications of potential reviewers for your manuscript. We have finally received three reviews for your manuscript. In general, all of them were quite positive. Still, there are some minor corrections that you need to take care of before the MS is formally accepted for publication in PLOS One. In practice, please consider it as accepted. Still, please solve the raised issues and resubmit it to us so it can be formally accepted. Overall, congratulations for your hard work!

Sincerely,

Daniel Silva

Reviewers' comments:

Reviewer's Responses to Questions

**Comments to the Author**

1. If the authors have adequately addressed your comments raised in a previous round of review and you feel that this manuscript is now acceptable for publication, you may indicate that here to bypass the “Comments to the Author” section, enter your conflict of interest statement in the “Confidential to Editor” section, and submit your "Accept" recommendation.

Reviewer #3: All comments have been addressed

Reviewer #4: (No Response)

Reviewer #5: All comments have been addressed

2. Is the manuscript technically sound, and do the data support the conclusions?

Reviewer #3: Yes

Reviewer #4: Yes

Reviewer #5: Yes

3. Has the statistical analysis been performed appropriately and rigorously? 

Reviewer #3: Yes

Reviewer #4: Yes

Reviewer #5: Yes

4. Have the authors made all data underlying the findings in their manuscript fully available?

Reviewer #3: Yes

Reviewer #4: Yes

Reviewer #5: Yes

5. Is the manuscript presented in an intelligible fashion and written in standard English?

Reviewer #3: Yes

Reviewer #4: Yes

Reviewer #5: Yes

6. Review Comments to the Author

Reviewer #3: (No Response)

Reviewer #4: A well-written paper that has addressed the questions raised by a previous reviewer and adds to the knowledge base in this field.

Reviewer #5: Dear authors, I congratulate you on this beautiful study. The topic is extremely relevant and important given the threats we face as a society for the conservation of biodiversity. In general, the text is very coherent with all the necessary elements to report the study and provide the necessary information. The analyses are well done and the results are organised and presented in a coherent manner. A few adjustments are relevant, both to conform to the journal's standards and to help the reader understand the text.

Abstract: The journal standard requires an abstract of up to 300 words. In the manuscript the abstract is 502. I suggest you look at https://chemistrycommunity.nature.com/posts/43071-how-to-write-an-abstract.

Line 293 has an incomplete sentence.

Line 453 gives a value of "0%". This definition looks strange in the text and it is better to say that there was no identification until species for invertebrates other than insects.

7. PLOS authors have the option to publish the peer review history of their article (what does this mean?). If published, this will include your full peer review and any attached files.

Reviewer #3: No

Reviewer #4: No

Reviewer #5: No

---

## [Author Response · Author response to Decision Letter 1]

1 Jun 2023

Response to Reviewers' Comments to the Author:

Reviewer #5: Dear authors, I congratulate you on this beautiful study. The topic is extremely relevant and important given the threats we face as a society for the conservation of biodiversity. In general, the text is very coherent with all the necessary elements to report the study and provide the necessary information. The analyses are well done and the results are organised and presented in a coherent manner. A few adjustments are relevant, both to conform to the journal's standards and to help the reader understand the text.

• Thank you for your positive comments on our paper and support of our work. Thank you also for your below suggestions, which have definitely improved it.

Abstract: The journal standard requires an abstract of up to 300 words. In the manuscript the abstract is 502. I suggest you look at https://chemistrycommunity.nature.com/posts/43071-how-to-write-an-abstract.

• Thank you for highlighting this. I have now cut the abstract (see tracked changes for lines 21-263) to abide by the journal’s word limit.

Line 293 has an incomplete sentence.

• Thank you for pointing this out. This was a result of changing the referencing style. I have now corrected this (see tracked changes in line 505).

Line 453 gives a value of "0%". This definition looks strange in the text and it is better to say that there was no identification until species for invertebrates other than insects.

• Thank you for suggesting this. I have incorporated this (see tracked changes in lines 665-666), and the sentence now reads much better.

---

## [Editor Report · Decision Letter 2]

5 Jun 2023

What can drawings tell us about children’s perceptions of nature?

PONE-D-22-22596R2

Dear Dr. Howlett,

We’re pleased to inform you that your manuscript has been judged scientifically suitable for publication and will be formally accepted for publication once it meets all outstanding technical requirements.

Kind regards,

Daniel de Paiva Silva, Ph.D.

Academic Editor

PLOS ONE

Additional Editor Comments (optional):

Dear Dr. Howlett,

I am pleased to infomr you that your manuscript has just been accepted for publication in PLoS One!

Congratulations on your hard work!

Sincerely,

Daniel Silva.
---

## [Editor Report · Acceptance letter]

8 Jun 2023

PONE-D-22-22596R2 

What can drawings tell us about children’s perceptions of nature? 

Dear Dr. Howlett:

I'm pleased to inform you that your manuscript has been deemed suitable for publication in PLOS ONE. Congratulations! Your manuscript is now with our production department. 

Kind regards, 

on behalf of

Dr. Daniel de Paiva Silva 

Academic Editor

PLOS ONE